# Stereoselective Synthesis of Multisubstituted Cyclohexanes by Reaction of Conjugated Enynones with Malononitrile in the Presence of LDA

**DOI:** 10.3390/molecules25245920

**Published:** 2020-12-14

**Authors:** Anastasiya V. Igushkina, Alexander A. Golovanov, Irina A. Boyarskaya, Ilya E. Kolesnikov, Aleksander V. Vasilyev

**Affiliations:** 1Department of Organic Chemistry, Institute of Chemistry, Saint Petersburg State University, Universitetskaya nab., 7/9, 199034 Saint Petersburg, Russia; kuznetsova.happy@yandex.ru (A.V.I.); iralbo@yahoo.com (I.A.B.); 2Department of Chemistry, Chemical Processes and Technologies, Togliatti State University, Belorusskaya ul., 14, 445667 Togliatti, Russia; aleksandgolovanov@yandex.ru; 3Center for Optical and Laser Materials Research, St. Petersburg State University, Ulyanovskaya ul., 5, 198504 Saint Petersburg, Russia; ilya.kolesnikov@spbu.ru; 4Department of Chemistry, Saint Petersburg State Forest Technical University, Institutsky per., 5, 194021 Saint Petersburg, Russia

**Keywords:** conjugated enynones, malononitrile, cyclohexanes, carbanionic reactions, photoluminescent compounds

## Abstract

Reaction of linear conjugated enynones, 1,5-diarylpent-2-en-4-yn-1-ones, with malononitrile in the presence of lithium diisopropylamide LDA, as a base, in THF at room temperature for 3–7 h resulted in the formation of the product of dimerization, multisubstituted polyfunctional cyclohexanes, 4-aryl-2,6-bis(arylethynyl)-3-(aryloxomethyl)-4-hydroxycyclohexane-1,1-dicarbonitriles, in yields up to 60%. Varying the reaction conditions by decreasing time and temperature and changing the ratio of starting compounds (enynone and malononitrile) allowed isolating some intermediate compounds, which confirmed a plausible reaction mechanism. The relative stability of possible stereoisomers of such cyclohexanes was estimated by quantum chemical calculations (DFT method). The obtained cyclohexanes were found to possess photoluminescent properties.

## 1. Introduction

Conjugated enynones are an important class of organic compounds due to the presence of several functional groups in their structures: double and triple carbon–carbon bonds, along with the carbonyl group. These enynones are starting reagents in many important organic transformations [1,2].

In our previous work, we studied a multicomponent reaction of linear conjugated enynones, 1,5-diarylpent-2-en-4-yn-1-ones, with malononitrile and sodium alkoxides affording substituted pyridines [3]. In that study [3], we showed just one example of the synthesis of multisubstituted cyclohexane by the reaction of 1,5-diphenylpent-2-en-4-yn-1-ones with malononitrile using lithium diisopropylamide (LDA), as a base, instead of sodium alkoxides. We decided to investigate this intriguing reaction deeper. Thus, the main goal of the current work was to study the reactions of 1,5-diarylpent-2-en-4-yn-1-ones **1a-i** (Figure 1) with malononitrile in the presence of LDA. We also tried to isolate some intermediate structures to elucidate the reaction mechanism.

Cyclohexane derivatives are essential components of many natural and synthetical biologically active products. For instance, such composite elements of volatile oils as terpenes have a cyclohexane skeleton. Terpenes and terpenoids have been found to be substances with multiple useful features [4]. They demonstrated anti-inflammatory [5], antifungal [6], antimicrobial [7,8,9], insecticidal [7,9,10], repellent [10,11], anticonvulsant [12], antioxidant [12,13], and even anticancer activities [8,13]. Derivatives of natural quinic acid play a role as antioxidants [14].

On the other hand, there are mamy synthetical biologically active substances having a cyclohexane core. It has been found that 2-acetyl-5-hydroxy-5-methyl-3-phenylcyclohexan-1-one and alkyl-4-hydroxy-4-methyl-2-oxo-6-phenylcyclohexane-1-carboxylates demonstrate antimicrobial activity [15]. Apart from that, methyl 2-cyanoacetylhydrazono-4-hydroxy-4-methyl-6-phenylcyclohexane-1-carboxilate possesses analgesic properties [16].

There are a few reported results on the reaction of chalcones (1,3-diarylprop-2-en-1-ones) with malononitrile [17,18,19,20,21,22]. Depending on conditions, several reaction products may be synthesized: Acyclic Michael addition product, substituted pyridines, aminoisophthalonitles or cyclohexanes. Cyclohexanes were synthesized by the reaction of chalcones with malononitrile using piperidine, as a base, in ethanol [17], in solvent-free conditions with sodium alkoxide [18], in water using hexadecyltrimethylammonium bromide [19], with potassium hydroxide in ethanol [20], in diethyl carbonate with quinine (cyclohexane was obtained as a minor reaction product) [21], and in water with tetrabutylammonium hydroxide (Scheme 1) [22]. In the last case, not only malononitrile but also ethyl cyanoacetate and nitromethane were used to get the corresponding cyclohexanes [22].

We note that in all described syntheses, the obtained cyclohexanes had identical stereochemistry of substituents in cyclohexane ring (Scheme 1) [18,21,22]. These cyclohexanes have four stereocenters, which means that theoretically, 16 stereoisomers can be obtained. However, only one pair of enantiomers of substituted cyclohexane was formed, and the reaction proceeds very diastereoselectively.

Thus, the reaction of 1,5-diarylpent-2-en-4-yn-1-ones **1a-i** with malononitrile promoted by LDA deserves to be investigated as a potential method of stereoselective synthesis of multisubstituted cyclohexanes bearing acetylenic substituents. In literature data, this type of ethynyl-substituted cyclohexanes is limitedly discussed. However, these compounds may have remarkable practical importance as photoluminescent complex ligands [23] and as useful building blocks for densely substituted cyclohexanes [24,25,26,27,28,29].

## 2. Results and Discussion

The results of reactions of enynones **1a-i** with malononitrile and LDA are presented in Table 1. Depending on reaction conditions, three types of products, target cyclohexanes **2** and intermediate compounds **3** and **4**, were obtained. Th structures of the obtained compounds were determined by NMR and HRMS (see Experimental part and Appendix A). Exact structures of compounds **2a**, **2b**, **2e** and **3a** were additionally confirmed by X-ray data (see Table 1, Experimental part and Appendix A). Pairs of enantiomers (racemates) of cyclohexanes **2** were obtained. Other diastereomers were not formed since we collected all compounds after chromatographic separation.

According to NOESY correlations (see spectra in Appendix A), the obtained cyclohexanes **2a**–**h** have identical relative configurations: two acetylenic substituents are oriented to one side of the cyclohexane ring while hydroxy and carbonyl groups are directed to the other side (Figure 2). In NOESY spectra, correlations between protons on one side of the cyclohexane ring are clearly observed. There are cross-peaks between H^a^ and H^b^, H^a^ and H^d^, H^a^ and OH, H^b^ and H^d^, H^c^ and H^e^, H^d^ and OH (Figure 2). NOESY correlations between protons on the opposite sides of the cyclohexane ring were not detected. This effect is caused by the fact that rotation around the cycle plane is limited.

There are some very specific signals of cyclohexane ring protons of **2** in ^1^H NMR (Figure 2). Proton H^a^ neighbor to CH_2_-group gives doublet of doublets in a range of 3.98–4.04 ppm in ^1^H NMR. Protons on the opposite sides of the cyclohexane ring, H^b^ and H^c^, are represented by doublets with constants about 12 Hz at 4.12–4.58 ppm. It is interesting that protons of the CH_2_ group give different signals, proton H^d^ on the same side of cyclohexane ring relatively to proton H^a^ is represented by a widened singlet at 5.22–5.23 ppm, while proton H^e^ on the opposite side to H^a^ gives a doublet with a small constant around 2 Hz at 2.38–2.40 ppm.

The formation of cyclohexane **2** takes place at room temperature for 3 h at a ratio of starting enynone **1**: malononitrile as 2:1 (Table 1, entries 1, 6–8, 10, 11, 13, 14). The maximum yield of target cyclohexane **2a** was reached in the reaction of diphenyl enynone **1a** (entry 1). Enynones bearing one substituent in the aromatic ring at carbonyl group (**1b**, **c**) or in the aromatic ring at triple bond (**1d**, **e**, **f**) gave target cyclohexanes **2** in lower yields (entries 6–8, 10, 11). An even more dramatic decrease of reaction product yield was observed when there were two substituents in aromatic rings of enynones **1 g**, **h** (entries 13, 14). In the series of substituted enynones, yields of **2** were higher in cases of electron-donating substituents in aromatic rings of enynones **1b**, **d**, **f** (entries 6, 8, 11). On the other hand, enynone **1i** bearing a very strong acceptor nitro group gave rise to a complex mixture of reaction products (entry 15). An increase in reaction time until 7 h led to a substantial decreasing in yields of cyclohexanes **2** (compare pairs of entries 8 and 9, 11 and 12). This may be caused by the destruction of compounds **2** under the basic reaction conditions.

In some cases, acyclic compounds **3**, as products of Michael addition of malononitrile to the double bond C=C of starting enynones **1**, were detected in reaction products along with cyclohexanes **2** (entries 8, 10, 11, 14). Compounds **3** are intermediate structures laying on a way of formation of **2** from **1**. We decided to find conditions for the preparation of compounds **3** in higher yields. For this purpose, an experiment using enynone **1a** and malononitrile with an equivalent ratio 1:1 at room temperature for 1 h was done; under these conditions, compound **3a** was obtained in a yield of 32% in a mixture with **2a** (entry 2). Decrease of reaction temperature to −40 °C led again to a mixture of **2a** and **3a** (entry 3). At lower temperature −70 °C for 0.5 h, a mixture of compound **3a** and diastereomers of dimeric diketone **4a**,**b** was obtained (entry 4). Prolongation of reaction till 3 h at −70 °C resulted in the formation of all three reaction products **2a**, **3a** and **4a**,**b** (entry 5). These data reveal that compounds **3** and **4** are intermediate compounds in the synthesis of **2**.

Michael addition product **3a** and diketone **4a**,**b** have very close chromatographic retention parameters. Thus, we were not able to individually isolate each of these substances by usual column chromatography on silica gel. However, preparative HPLC separation of **3a** and **4a**,**b** was successful in getting these compounds. Diastereomers **4a**,**b** may be discerned by NMR.

A plausible reaction mechanism is presented in Scheme 2. The Michael addition of malononitrile anion to the carbon–carbon double bond of enynone **1** gives rise to anion **A**, which is isomerized into anion **B**. Aqueous work up at this reaction step affords compound **3**. Anion **B** may react with one more molecule of **1** affording anion **C**, and this stage goes stereoselectively. Work up to at this stage provides compound **4**. Cyclization of anion **C** into species **D** forms cyclohexane ring. This stage proceeds very stereoselectively, furnishing only one stereoisomer of **2**. Finally, aqueous workup of anion **D** results in the formation of compound **2**.

DFT calculations of relative energies of 8 enantiomeric pairs of compound **2a** showed that the obtained isomer **2a** had minimal energy. The difference in relative energies of diastereomers reaches up to 49 kJ/mol (see Appendix A). In the most stable isomer, all the bulky substituents, except the OH group, take equatorial positions. There is an additional stabilization of the structure by hydrogen bonding between the carbonyl oxygen and a hydroxyl group (Figure 3).

Additionally, photoluminescent properties of the obtained cyclohexanes **2** were investigated. It was found that upon excitation at 436–465 nm, maximums of emission were observed at 370–550 nm. Compound **2a** having four phenyl rings, showed the maximum of emission at 365 nm upon the excitation at 446 nm, while compounds **2b**, **2c**, **2e,** having substituted aromatic rings, revealed the maximums of emission at 520–550 nm upon the excitation at 436–465 nm (see details in SI). Quantum yields of photoluminescence for **2a**, **2b** and **2c** were 1.36%, 4.95% and 2.81%, respectively. We suggested that quantum yields were low because of the flexibility of the saturated cyclohexane cycle. Cyclohexanes **2** have many degrees of freedom, and a significant part of the absorbed energy is spent on vibrations. This problem theoretically may be solved by coordination with a metal ion. Compounds **2** have several groups, CN, CO, OH, that are typical for aggregation-induced emission-based sensors for low concentration toxic ion detection [30]. It may be a perspective direction for further research.

## 3. Conclusions

We developed a synthesis of multisubstituted polyfunctional cyclohexanes, bearing two cyano groups and two arylethynyl substituents, by the reaction of conjugated 1,5-diarylpent-1-en-4-yn-1-ones with malononitrile in the presence of a strong base as LDA. This is a stereoselective dimerization leading to the formation of only one diastereomer of such cyclohexanes. Varying reaction conditions (ratio of starting compounds, reaction temperature and time), we were able to isolate some intermediate compounds laying on the reaction pathway to target cyclohexanes. That shed light on the proposed reaction mechanism. The obtained cyclohexanes were found to show photoluminescent properties.

### 3.1. Experimental Section

The NMR spectra of solutions of compounds in CDCl_3_ were recorded on Bruker AVANCE III 400 spectrometer [at 400, 100 and 61 MHz for ^1^H, ^13^C NMR spectra, respectively] at 25 °C. The solvent residual signals CDCl_3_ (δ 7.26 ppm) for ^1^H NMR spectra, the carbon signal of CDCl_3_ (δ 77.0 ppm) for ^13^C NMR spectra. IR spectra of compounds were taken with a Bruker spectrometer. HRMS was carried out on a Bruker maXis HRMS-ESI-QTOF instrument. Photoluminescence properties were measured on Fluorolog-3 (Horiba) spectrofluorimeter at room temperature. Steady-state measurements were performed using Xe-arc lamp (450 W) as an excitation source with slits spectral width of 3 nm. LED (370 nm, pulse duration of 1.2 ns) was used to carry out lifetime measurements. Quantum yield was obtained through the absolute technique using an integrating sphere (Quanta-phi, 6 inches). The preparative reactions were monitored by thin-layer chromatography carried out on silica gel plates (Alugram SIL G/UV-254), using a UV light for detection. Preparative TLC was performed on silica gel Chemapol L 5/40, respectively. HPLC was done on a Waters machine using gradient elution with acetonitrile-water mixtures.

### 3.2. X-ray Diffraction Study 

Single crystal X-ray analysis was performed at single crystal diffractometer Agilent Technologies (Oxford diffraction) “Supernova”. The crystal was kept at 100(2) K during data collection. Using Olex2 [31], the structure was solved with the ShelXS [32] structure solution program using Direct Methods and refined with the ShelXL refinement package using least-squares minimization. CCDC 1998357–(**2b**), CCDC 1998289–(**2e**), CCDC 1998291–(**3a**) contain the supplementary crystallographic data (see Appendix A), which can be obtained free of charge at www.ccdc.cam.ac.uk/conts/retrieving.html or from the Cambridge Crystallographic Data Centre, 12 Union Road, Cambridge CB2 1EZ, UK; Fax: +44-1223-336-033; E-mail: deposit@ccdc.cam.ac.uk.

### 3.3. DFT Calculations

All computations were carried out at the DFT/HF hybrid level of theory using hybrid exchange functional M06 by using GAUSSIAN 2009 program packages [33]. The geometries optimization was performed using the M06/6-311+G(2d,2p) basis set (standard 6-311 basis set added with polarization (d, p) and diffuse functions) using water as a solvent. Optimizations were performed on all degrees of freedom, and solvent-phase optimized structures were verified as true minima with no imaginary frequencies. The Hessian matrix was calculated analytically for the optimized structures in order to prove the location of the correct minima and to estimate the thermodynamic parameters. Solvent-phase calculations used the polarizable continuum model (PCM).

Preparation and characterization of starting enynones **1** has been previously described [34,35].

The general procedure for the synthesis of 4-aryl-2,6-bis(aryletynyl)-4-hydroxy-3-(aryloxomethyl)cyclohexane-1,1-dicarbonitriles (**2a**–**h**) and 5-aryl-3-arylethynyl-1-cyano-5-oxopentannitriles (**3a**,**d**–**f**,**h**) from 1,5-diarylpent-2-en-4-yn-1-ones (**1a**–**h**). A flask with 2 equiv. of enynone 1 and 1 equiv. of malononitrile was filled with argon, then 2 mL of absolute THF was added and 1 equiv. LDA in THF solution (2 mmol/mL) was dropwise added, and the mixture was stirring at room temperature for 3–7 h. The reaction mixture was quenched with 0.5 mL of NH_4_Cl solution and was extracted with CH_2_Cl_2_ (3 × 30 mL). The combined organic layers were dried with Na_2_SO_4_. The solvent was evaporated under the reduced pressure. The obtained residue was subjected to preparative TCL on silica gel with elution by petroleum ether-ethyl acetate mixture in the ratio 80:20 to get pure compounds 2 and 3.

*(2RS,3SR,4RS,6SR)-4-Hydroxy-3-(oxophenylmethyl)-4-phenyl-2,6-bis(phenyletynyl)cyclohexane-1,1-dicarbonitrile* (**2a**) [3]. Obtained from **1a** (100 mg, 0.44 mmol) in a yield of 70 mg (60%). Obtained from **1a** (50 mg, 0.22 mmol) and malononitrile 14.5 mg (0.22 mmol) in a yield of 26 mg (45%). Obtained from **1a** (50 mg, 0.22 mmol) and malononitrile 14.5 mg (0.22 mmol) at −40 °C in a yield of 28 mg (48%). Obtained from **1a** (50 mg, 0.22 mmol) and malononitrile 14.5 mg (0.22 mmol) at −70 °C in a yield of 10 mg (17%). Colorless solid. ^1^H NMR, δ, ppm: 2.40 s (1H, OH), 2.42–2.43 m (1H, CH_2_), 4.06 dd (1H, CH-CH_2_, *J* = 5.9, 9.9 Hz), 4.16 d (1H, CH-CH-CO, *J* = 11.6 Hz), 4.60 d (1H, CH-CO, *J* = 11.5 Hz), 5.24 s (1H, CH_2_), 6.97 d (2H, H^Ph^, *J* = 7.3 Hz), 7.13–7.19 m (3H), 7.22–7.27 m (3H), 7.35–7.47 m (7H), 7.52–7.57 m (3H), 7.88 d (2H, H^Ph^, *J* = 7.7 Hz). ^13^C NMR, δ, ppm: 34.3 (CH-CH_2_), 37.3 (CH_2_), 42.0 (C-(CN)_2_), 46.2 (CH-CH-CO), 49.9 (CH-CO), 73.8 (C-OH), 82.2 (C≡), 83.9 (C≡), 86.7 (C≡), 89.4 (C≡), 112.4 (CN), 113.9 (CN), 120.9, 121.7, 124.4, 128.0, 128.1, 128.3, 128.66, 128.71, 128.8, 129.0, 129.1, 131.7, 132.1, 134.4, 137.2, 142.8, 202.8 (CO). IR (KBr): 1662 cm^−1^ (CO), 2239 cm^−1^ (CN, C≡C), 3442 cm^−1^ (OH…O). HRMS (ESI-TOF) *m/z*: [M + Na]^+^ Calcd for C_37_H_26_N_2_O_2_Na 553.1892; Found 553.1886. X-ray of **2a** were given previously [3].

*(2RS,3SR,4RS,6SR)-4-Hydroxy-4-(4-methylphenyl)-3-(4-methylphenyl)oxomethyl-2,6-bis(phenyletynyl)cyclohexane-1,1-dicarbonitrile* (**2b**). Obtained from **1b** (45 mg, 0.18 mmol) in a yield of 20 mg (40%). Colorless solid. ^1^H NMR, δ, ppm: 2.23 s (3H, CH_3_), 2.35–2.37 m (2H, OH, CH_2_), 2.38 s (3H, CH_3_), 4.01–4.06 m (1H, CH-CH_2_), 4.11 d (1H, CH-CH-CO, *J* = 11.5 Hz), 4.54 d (1H, CH-CO, *J* = 11.5 Hz), 5.33 s (1H, CH_2_), 6.93 d (2H, H^Ar^, *J* = 7.1 Hz), 7.04 d (2H, H^Ar^, *J* = 8.0 Hz), 7.15–7.21 m (4H, H^Ar^), 7.25–7.27 m (1H, H^Ar^), 7.30–7.36 m (5H, H^Ar^), 7.52 dd (2H, H^Ar^, *J* = 2.0, 7.5 Hz), 7.80 d (2H, H^Ar^, *J* = 8.3 Hz). ^13^C NMR, δ, ppm: 21.7 (CH_3_), 28.3 (CH_3_), 34.3 (CH-CH_2_), 37.4 (CH_2_), 42.2 (C-(CN)_2_), 46.2 (CH-CH-CO), 49.4 (CH-CO), 73.6 (C-OH), 82.4 (C≡), 84.0 (C≡), 86.6 (C≡), 89.3 (C≡), 112.4 (CN), 114.0 (CN), 121.0, 121.8, 124.3, 128.0, 128.3, 128.9, 129.0, 129.1, 129.3, 129.4, 131.7, 132.1, 134.8, 137.6,140.2, 145.6, 202.2 (CO). IR (KBr): 1659 cm^−1^ (CO), 2238 cm^−1^ (CN, C≡C), 3432 cm^−1^ (OH…O). HRMS (ESI-TOF) *m/z*: [M + Na]^+^ Calcd for C_39_H_30_N_2_O_2_Na 581.2205; Found 581.2199.

*(2RS,3SR,4RS,6SR)-4-Hydroxy-4-(4-metoxylphenyl)-3-(4-methoxyphenyl)oxomethyl-2,6-bis(phenyletynyl)cyclohexane-1,1-dicarbonitrile* (**2c**). Obtained from **1c** (45 mg, 0.17 mmol) in a yield of 13 mg (26%). Colorless solid. ^1^H NMR, δ, ppm: 2.34–2.36 m (2H, OH, CH_2_), 3.72 s (3H, OCH_3_), 3.84 s (3H, OCH_3_), 4.01–4.06 m (1H, CH-CH_2_), 4.11 d (1H, CH-CH-CO, J = 11.5 Hz), 4.47 d (1H, CH-CO, J = 11.5 Hz), 5.44 s (1H, CH_2_), 6.76 d (2H, H^Ar^, J = 8.8 Hz), 6.87 d (2H, H^Ar^, J = 8.9 Hz), 6.99 d (2H, H^Ar^, J = 7.2 Hz), 7.18 t (2H, H^Ar^, J = 7.6 Hz), 7.26–7.29 m (1H, H^Ar^), 7.32–7.38 m (5H, H^Ar^), 7.52 dd (2H, H^Ar^, J = 2.0, 7.5 Hz), 7.90 d (2H, H^Ar^, J = 8.9 Hz). ^13^C NMR, δ, ppm: 34.3 (CH-CH_2_), 37.4 (CH_2_), 42.3 (C-(CN)_2_), 46.2 (CH-CH-CO), 49.3 (CH-CO), 51.2 (OCH_3_), 55.6 (OCH_3_), 73.4 (C-OH), 82.5 (C≡), 84.1 (C≡), 86.5 (C≡), 89.2 (C≡), 112.4 (CN), 113.9, 114.0, 121.1, 121.8, 125.6, 128.1, 128.3, 128.9, 129.0, 130.2, 131.5, 131.7, 132.1, 135.3, 159.0 (C^Ar^-O), 164.7 (C^Ar^-O), 200.6 (CO). IR (KBr): 1650 cm^−1^ (CO), 2238 cm^−1^ (CN, C≡C), 3418 cm^−1^ (OH…O). HRMS (ESI-TOF) m/z: [M + Na]^+^ Calcd for C_39_H_30_N_2_O_4_Na 613.2103; Found 613.2098.

*(2RS,3SR,4RS,6SR)-4-Hydroxy-2,6-bis((4-methylphenyl)etynyl)-3-(oxophenylmethyl)-4-phenylcyclohexane-1,1-dicarbonitrile* (**2d**). Obtained from **1d** (50 mg, 0.20 mmol), reaction time—3 h in a yield of 30 mg (54%). Obtained from **1d** (50 mg, 0.20 mmol), reaction time—7 h in a yield of 14 mg (25%). Colorless solid. ^1^H NMR, δ, ppm: 2.30 s (3H, CH_3_), 2.35–2.40 m (2H, OH, CH_2_), 2.38 s (3H, CH_3_), 4.04 dd (1H, CH-CH_2_, *J* = 5.9, 9.9 Hz), 4.13 d (1H, CH-CH-CO, *J* = 11.5 Hz), 4.58 d (1H, CH-CO, *J* = 11.5 Hz), 5.23 s (1H, CH_2_), 6.85 d (2H, H^Ar^, *J* = 8.0 Hz), 6.98 d (2H, H^Ar^, *J* = 8.0 Hz), 7.16 d (2H, H^Ar^, *J* = 8.1 Hz), 7.23 t (2H, H^Ar^, *J* = 7.6 Hz), 7.37–7.44 m (7H, H^Ar^), 7.54 t (1H, H^Ar^, *J* = 7.4 Hz), 7.87 d (2H, H^Ar^, *J* = 7.8 Hz). ^13^C NMR, δ, ppm: 21.45 (CH_3_), 21.52 (CH_3_), 34.3 (CH-CH_2_), 37.3 (CH_2_), 42.0 (C-(CN)_2_), 46.4 (CH-CH-CO), 50.0 (CH-CO), 73.8 (C-OH), 81.5 (C≡), 83.3 (C≡), 86.8 (C≡), 89.5 (C≡), 112.4 (CN), 114.0 (CN), 117.8, 118.7, 124.4, 127.9, 128.6, 128.7, 128.78, 128.83, 129.1, 131.6, 132.0, 134.3, 137.2, 139.1, 139.3, 142.9, 202.9 (CO). IR (KBr): 1687 cm^−1^ (CO), 2231 cm^−1^ (CN, C≡C), 3433 cm^−1^ (OH…O). HRMS (ESI-TOF) *m/z*: [M + Na]^+^ Calcd for C_39_H_30_N_2_O_2_Na 581.2205; Found 581.2199.

*(2RS,3SR,4RS,6SR)-2,6-Bis((4-chlorophenyl)etynyl)-4-hydroxy-3-(oxophenylmethyl)-4-phenylcyclohexane-1,1-dicarbonitrile* (**2e**). Obtained from **1e** (50 mg, 0.19 mmol) in a yield of 13 mg (23%). Colorless solid. ^1^H NMR, δ, ppm: 2.38 s (1H, OH), 2.40 d (1H, CH_2_, *J* = 2.0 Hz), 4.04 dd (1H, CH-CH_2_, *J* = 6.9, 8.9 Hz), 4.13 d (1H, CH-CH-CO, *J* = 11.5 Hz), 4.57 d (1H, CH-CO, *J* = 11.5 Hz), 5.22 s (1H, CH_2_), 6.86 d (2H, H^Ar^, *J* = 8.6 Hz), 7.16 dd (3H, H^Ar^, *J* = 2.8, 8.0 Hz), 7.22–7.26 m (2H, H^Ar^), 7.33 d (2H, H^Ar^, *J* = 8.6 Hz), 7.38–7.45 m (7H, H^Ar^), 7.85 d (2H, H^Ar^, *J* = 7.3 Hz). ^13^C NMR, δ, ppm: 34.3 (CH-CH_2_), 37.3 (CH_2_), 41.8 (C-(CN)_2_), 45.9 (CH-CH-CO), 49.7 (CH-CO), 73.7 (C-OH), 83.1 (C≡), 84.8 (C≡), 86.6 (C≡), 88.3 (C≡), 112.2 (CN), 113.7 (CN), 119.2, 120.1, 124.3, 125.5, 128.0, 128.2, 128.5, 128.70, 128.74, 132.8, 133.3, 134.5, 135.2, 135.4, 137.1, 142.7, 202.5 (CO). IR (KBr): 1664 cm^−1^ (CO), 2243 cm^−1^ (CN, C≡C), 3437 cm^−1^ (OH…O). HRMS (ESI-TOF) *m/z*: [M + Na]^+^ Calcd for C_37_H_24_Cl_2_N_2_O_2_Na 621.1113; Found 621.1107.

*(2RS,3SR,4RS,6SR)-4-Hydroxy-2,6-bis((4-methoxyphenyl)etynyl)-3-(oxophenylmethyl)-4-phenylcyclohexane-1,1-dicarbonitrile* (**2f**). Obtained from **1f** (50 mg, 0.19 mmol), reaction time—3 h in a yield of 18 mg (33%). Obtained from **1f** (47 mg, 0.18 mmol), reaction time—7 h in a yield of 7 mg (13%). Colorless solid. ^1^H NMR, δ, ppm: 2.37–2.40 m (2H, OH, CH_2_), 3.77 s (3H, CH_3_), 3.84 s (3H, CH_3_), 4.02 dd (1H, CH-CH_2_, *J* = 6.0, 9.8 Hz), 4.12 d (1H, CH-CH-CO, *J* = 11.5 Hz), 4.57 d (1H, CH-CO, *J* = 11.5 Hz), 5.23 s (1H, CH_2_), 6.69 d (2H, H^Ar^, *J* = 8.9 Hz), 6.84–6.91 m (4H, H^Ar^), 7.23 t (2H, H^Ar^, *J* = 7.6 Hz), 7.37–7.46 m (6H, H^Ar^), 7.50–7.55 m (2H, H^Ar^), 7.91 d (2H, H^Ar^, *J* = 7.2 Hz). ^13^C NMR (selected signals, obtained from spectrum of mixture with unknown compounds), δ, ppm: 34.3 (CH-CH_2_), 37.4 (CH_2_), 42.0 (C-(CN)_2_), 46.5 (CH-CH-CO), 50.0 (CH-CO), 55.30 (OCH_3_), 55.31 (OCH_3_), 73.8 (C-OH), 80.9 (C≡), 82.6 (C≡), 86.6 (C≡), 89.4 (C≡), 112.5 (CN), 113.0 (CN), 202.9 (CO). IR (KBr): 1664 cm^−1^, 1683 cm^−1^ (CO), 2230 cm^−1^ (CN, C≡C), 3434 cm^−1^ (OH…O). HRMS (ESI-TOF) *m/z*: [M + Na]^+^ Calcd for C_39_H_30_N_2_O_4_Na 613.2103; Found 613.2098.

*(2RS,3SR,4RS,6SR)-4-(4-Bromophenyl)-3-(4-bromphenyl)oxomethyl-4-hydroxy-2,6-bis(4-methylphenyl)etynylcyclohexane-1,1-dicarbonitrile* (**2 h**). Obtained from **1 h** (50 mg, 0.15 mmol) in a yield of 9 mg (17%). Colorless solid. ^1^H NMR, δ, ppm: 2.33 s (3H, CH_3_), 2.34–2.37 m (2H, OH, CH_2_), 2.38 s (3H, CH_3_), 4.00 dd (1H, CH-CH_2_, J = 4.9, 10.9 Hz), 4.08 d (1H, CH-CH-CO, J = 11.5 Hz), 4.45 d (1H, CH-CO, J = 11.5 Hz), 5.21 d (1H, CH_2_, J = 2.2 Hz), 6.86 d (2H, H^Ar^, J = 8.1 Hz), 7.02 d (2H, H^Ar^, J = 8.0 Hz), 7.16 d (2H, H^Ar^, J = 8.0 Hz), 7.29 d (2H, H^Ar^, J = 8.6 Hz), 7.39 dd (2H, H^Ar^, J = 6.1, 8.2 Hz), 7.56 d (2H, H^Ar^, J = 8.6 Hz), 7.74 d (2H, H^Ar^, J = 8.6 Hz). ^13^C NMR, δ, ppm: 21.48 (CH_3_), 21.52 (CH_3_), 34.2 (CH-CH_2_), 37.3 (CH_2_), 42.0 (C-(CN)_2_), 46.3 (CH-CH-CO), 49.6 (CH-CO), 73.6 (C-OH), 81.3 (C≡), 82.9 (C≡), 87.0 (C≡), 90.1 (C≡), 112.3 (CN), 113.7 (CN), 117.5, 118.5, 122.2, 126.1, 129.0, 129.1, 129.8, 130.2, 131.5, 131.89, 131.94, 132.2, 135.7, 139.3, 139.7, 142.1, 201.7 (CO). IR (KBr): 1664 cm^−1^ (CO), 2235 cm^−1^ (CN, C≡C), 3437 cm^−1^ (OH…O). HRMS (ESI-TOF) m/z: [M + Na]^+^ Calcd for C_39_H_28_Br_2_N_2_O_2_Na 737.0415; Found 737.0410.

*(2RS,3SR,4RS,6SR)-4-(4-Chlorophenyl)-3-(4-chlorophenyl)oxomethyl-4-hydroxy-2,6-bis(4-methylphenyl)etynylcyclohexane-1,1-dicarbonitrile* (**2 g**). Obtained from **1 g** (50 mg, 0.18 mmol) in a yield of 10 mg (18%). Colorless solid. ^1^H NMR, δ, ppm: 3.63 ddd (2H, CH_2_, J = 6.7, 18.5, 27.3 Hz), 3.97 dt (1H, CH-C≡C, J = 4.6, 9.1 Hz), 4.61 d (1H, CH-(CN)_2_, J = 4.7 Hz), 7.32–7.44 m (2H, H^Ar^), 7.46–7.49 m (2H, H^Ar^), 7.55 t (2H, H^Ar^, J = 7.8 Hz), 7.68 t (1H, H^Ar^, J = 6.8 Hz), 8.01 d (2H, H^Ar^, J = 7.2 Hz). ^13^C NMR, δ, ppm: 21.48 (CH_3_), 21.52 (CH_3_), 34.2 (CH-CH_2_), 37.3 (CH_2_), 42.0 (C-(CN)_2_), 46.3 (CH-CH-CO), 49.8 (CH-CO), 73.6 (C-OH), 81.3 (C≡), 82.9 (C≡), 87.0 (C≡), 90.0 (C≡), 112.3 (CN), 113.7 (CN), 117.5, 118.5, 125.8, 128.9, 129.0, 129.1, 129.2, 130.1, 131.5, 131.9, 134.0, 135.3, 139.3, 139.7, 141.49, 141.53, 201.4 (CO). IR (KBr): 1664 cm^−1^ (CO), 2237 cm^−1^ (CN, C≡C), 3440 cm^−1^ (OH…O). HRMS (ESI-TOF) m/z: [M − H]^-^ Calcd for C_39_H_27_Cl_2_N_2_O_2_ 625.1450; Found 625.1444.

*1-Cyano-5-phenyl-3-phenylethynyl-5-oxopentannitrile* (**3a**) [3]. Obtained from **1a** (50 mg, 0.22 mmol) and malononitrile 14.5 mg (0.22 mmol) in a yield of 21 mg (32%). Obtained from **1a** (50 mg, 0.22 mmol) and malononitrile 14.5 mg (0.22 mmol) at −40 °C in a yield of 35 mg (53%). Obtained from **1a** (50 mg, 0.22 mmol) and malononitrile 14.5 mg (0.22 mmol) at −70 °C, 0.5 h in a yield of 14 mg (27%). Obtained from **1a** (50 mg, 0.22 mmol) and malononitrile 14.5 mg (0.22 mmol) at −70 °C, 3 h in a yield of 15 mg (23%). Colorless solid. ^1^H NMR, δ, ppm: 3.57 dd (1H, CH_2_, *J* = 8.8, 18.5 Hz), 3.69 dd (1H, CH_2_, *J* = 4.6, 18.5 Hz), 3.98 dt (1H, CH-C≡C, *J* = 4.6, 9.1 Hz), 4.61 d (1H, CH-(CN)_2_, *J* = 4.7 Hz), 7.33–7.39 m (3 H, H^Ar^), 7.49–7.53 m (2H, H^Ar^), 7.56 d (2H, H^Ar^, *J* = 7.9 Hz), 7.67 t (1H, H^Ar^, *J* = 7.4 Hz), 8.02 d (2H, H^Ar^, *J* = 7.3 Hz).^13^C NMR, δ, ppm: 28.0 (CH-(CN)_2_), 30.6 (CH-C≡C), 40.3 (CH_2_), 83.4 (C≡), 86.9 (C≡), 111.0 (CN), 111.6 (CN), 121.4, 128.2, 128.4, 129.0, 129.1, 132.0, 134.4, 135.5, 195.9 (CO). IR (KBr): 1685 cm^−1^ (CO), 2236 cm^−1^, 2258 cm^−1^ (CN, C≡C). HRMS (ESI-TOF) *m/z*: [M + Na]^+^ Calcd for C_20_H_14_N_2_ONa 321.1004; Found 321.0998.

*1-Cyano-3-(4-methylphenyl)ethynyl-5-oxo-5-phenylpentannitrile* (**3d**). Obtained from **1d** (50 mg, 0.20 mmol), reaction time—3 h in a yield of 15 mg (25%). Obtained from **1d** (50 mg, 0.20 mmol), reaction time—7 h in a yield of 14 mg (23%). Colorless solid. ^1^H NMR, δ, ppm: 2.38 s (3H, CH_3_), 3.57 dd (1H, CH_2_, *J* = 8.8, 18.5 Hz), 3.69 dd (1H, CH_2_, *J* = 4.5, 18.5 Hz), 3.96 dt (1H, CH-C≡C, *J* = 4.6, 9.0 Hz), 4.60 d (1H, CH-(CN)_2_, *J* = 4.7 Hz), 7.16 d (2H, H^Ar^, *J* = 8.0 Hz), 7.39 d (2H, H^Ar^, *J* = 8.1 Hz), 7.54 t (2H, H^Ph^, *J* = 7.8 Hz), 7.67 t (1H, H^Ph^, *J* = 7.4 Hz), 8.02 d (2H, H^Ph^, *J* = 7.2 Hz). ^13^C NMR, δ, ppm: 21.5 (CH_3_), 28.0 (CH-(CN)_2_), 30.6 (CH-C≡C), 40.3 (CH_2_), 82.7 (C≡), 87.1 (C≡), 111.0 (CN), 111.6 (CN), 118.4, 128.2, 129.0, 129.1, 131.9, 134.4, 135.5, 139.4, 197.0 (CO). IR (KBr): 1682 cm^−1^ (CO), 2211 cm^−1^ (CN, C≡C). HRMS (ESI-TOF) *m/z*: [M + Na]^+^ Calcd for C_21_H_16_N_2_ONa 335.1160; Found 335.1155.

*3-(4-Chlorophenyl)ethynyl-1-cyano-5-oxo-5-phenylpentannitrile* (**3e**). Obtained from **1e** (50 mg, 0.19 mmol) in a yield of 10 mg (16%). Colorless solid. ^1^H NMR, δ, ppm: 3.57 dd (1H, CH_2_, *J* = 8.8, 18.5 Hz), 3.68 dd (1H, CH_2_, *J* = 4.6, 18.5 Hz), 3.97 dt (1H, CH-C≡C, *J* = 4.6, 9.1 Hz), 4.61 d (1H, CH-(CN)_2_, *J* = 4.7 Hz), 7.32–7.44 m (2H, H^Ar^), 7.46–7.49 m (2H, H^Ar^), 7.55 t (2H, H^Ar^, *J* = 7.8 Hz), 7.68 t (1H, H^Ar^, *J* = 6.8 Hz), 8.01 d (2H, H^Ar^, *J* = 7.2 Hz). ^13^C NMR, δ, ppm: 28.3 (CH-(CN)_2_), 30.6 (CH-C≡C), 40.1 (CH_2_), 84.4 (C≡), 85.8 (C≡), 110.9 (CN), 111.5 (CN), 199.9, 124.5, 125.5, 128.2, 128.8, 129.0, 133.3, 134.5, 195.8 (CO). IR (KBr): 1685 cm^−1^ (CO), 2216 cm^−1^ (CN, C≡C). HRMS (ESI-TOF) *m/z*: [M + Na]^+^ Calcd for C_20_H_13_N_2_ONa 355.0614; Found 355.0609.

*1-Cyano-3-(4-methoxyphenyl)ethynyl-5-oxo-5-phenylpentannitrile* (**3f**). Obtained from **1f** (50 mg, 0.19 mmol), reaction time—3 h in a yield of 5 mg (8%). Obtained from **1f** (47 mg, 0.18 mmol), reaction time—7 h in a yield of 1 mg (2%). Colorless solid. ^1^H NMR, δ, ppm: 3.56 dd (1H, CH_2_, *J* = 8.8, 18.5 Hz), 3.68 dd (1H, CH_2_, *J* = 4.6, 18.5 Hz), 3.84 s (3H, OCH_3_), 3.96 dt (1H, CH-C≡C, *J* = 4.6, 9.0 Hz), 4.60 d (1H, CH-(CN)_2_, *J* = 4.7 Hz), 6.87 d (2H, H^Ar^, *J* = 8.9 Hz), 7.43 d (2H, H^Ar^, *J* = 8.8 Hz), 7.54 t (2H, H^Ph^, *J* = 7.7 Hz), 7.67 t (1H, H^Ph^, *J* = 7.4 Hz), 8.02 d (2H, H^Ph^, *J* = 7.2 Hz).^13^C NMR, δ, ppm: 28.1 (CH-(CN)_2_), 30.6 (CH-C≡C), 40.4 (CH_2_), 55.3 (OCH_3_), 82.0 (C≡), 86.9 (C≡), 111.1 (CN), 111.7 (CN), 113.5, 114.0, 128.2, 129.0, 133.5, 134.3, 135.5, 160.2 (C^Ar^-O), 196.0 (CO). IR (KBr): 1680 cm^−1^ (CO), 2231 cm^−1^ (CN, C≡C). HRMS (ESI-TOF) *m/z*: [M + Na]^+^ Calcd for C_21_H_16_N_2_O_2_Na 351.1109; Found 351.1104.

*5-(4-Bromophenyl)-1-cyano-3-(4-methylphenyl)ethynyl-5-oxopentannitrile* (**3 h**). Obtained from **1 h** (50 mg, 0.15 mmol) in a yield of 3 mg (5%). Colorless solid. ^1^H NMR, δ, ppm: 2.38 s (3H, CH_3_), 3.52 dd (1H, CH_2_, *J* = 8.7, 18.5 Hz), 3.64 dd (1H, CH_2_, *J* = 4.7, 18.5 Hz), 3.95 dt (1H, CH-C≡C, *J* = 4.7, 9.1 Hz), 4.57 d (1H, CH-(CN)_2_, *J* = 4.7 Hz), 7.16 d (2H, H^Ar^, *J* = 7.9 Hz), 7.37 d (2H, H^Ar^, *J* = 8.1 Hz), 7.69 d (2H, H^Ar^, *J* = 8.6 Hz), 7.88 d (2H, H^Ar^, *J* = 8.6 Hz). ^13^C NMR, δ, ppm: 21.5 (CH_3_), 28.0 (CH-(CN)_2_), 30.5 (CH-C≡C), 40.3 (CH_2_), 82.4 (C≡), 87.3 (C≡), 111.0 (CN), 111.5 (CN), 118.2, 129.1, 129.7, 129.8, 131.9, 132.4, 134.2, 139.5, 195.0 (CO). IR (KBr): 1689 cm^−1^ (CO), 2231 cm^−1^ (CN, C≡C). HRMS (ESI-TOF) *m/z*: [M + Na]^+^ Calcd for C_21_H_15_BrN_2_ONa 413.0265; Found 413.0260.

Procedure for the synthesis of *(R,S)-2-cyano-5-oxo-2-(3-oxo-3-phenyl-1-phenylmethylpropyl)-5-phenyl-3-phenylethylpentannitrile* (**4a**,**b**). The (*E*)-1,5-diphenylpent-2-en-4-yn-1-one 1a (50 mg, 0.22 mmol) in 2 mL of absolute THF and malononitrile (14.5 mg, 0.22 mmol) in 1 mL of absolute THF were added to a flask filled with argon. The flask was placed in a liquid nitrogen–acetone bath at −70 °C, the solution (0.11 mL, 2 mmol/mL) of LDA (0.22 mmol) in THF was dropwise added, and the mixture was stirring at −70 °C for 0.5 h (11 mg of unreacted starting enynone 1a was obtained) or 3 h (no unreacted starting enynone 1a was fixed). The reaction mixture was washed with 0.5 mL of NH_4_Cl solution and was extracted with CH_2_Cl_2_ (3 × 30 mL). The combined organic layers were dried with Na_2_SO_4_. The solvent was evaporated under the reduced pressure. The obtained residue was subjected to preparative TCL on silica gel with elution by petroleum ether-ethyl acetate mixture in the ratio 80:20 to get pure compounds. The fraction with a mixture of 3a and 4a,b was separated using a preparative HPLC.

*(R,S)-2-Cyano-5-oxo-2-(3-oxo-3-phenyl-1-phenylmethylpropyl)-5-phenyl-3-phenylethylpentannitrile* (**4a,b**). *Mixture of diastereomers.* Obtained from **1a** (39 mg, 0.17 mmol) in a yield of **4a** 5 mg (11%), yield of **4b** 7 mg (16%). Obtained from **1a** (50 mg, 0.22 mmol) in a yield of **4a** 12 mg (20%), yield of **4b** 6 mg (11%). Colorless solid. ^1^H NMR, **4a** (selected signals, obtained from spectrum of mixture of isomers), δ, ppm: 3.64 dd (2H, CH_2_, *J* = 3.5, 16.8 Hz), 3.91 dd (2H, CH_2_, *J* = 10.0, 16.8 Hz), 4.57 dd (2H, CH, *J* = 3.4, 10.0 Hz), 7.65 t (2H, H^Ph^, *J* = 7.4 Hz). ^1^H NMR, **4b** (selected signals, obtained from spectrum of mixture of isomers), δ, ppm: 3.56 dd (2H, CH_2_, *J* = 3.2, 16.5 Hz), 3.88 dd (2H, CH_2_, *J* = 10.3, 16.5 Hz), 4.39 dd (2H, CH, *J* = 3.2, 10.2 Hz). ^1^H NMR, (signals of mixture of isomers) δ, ppm: 7.29–7.36 m, 7.43–7.48 m, 7.51–7.56 m, 8.06–8.08 m. ^13^C NMR, **4a** (selected signals, obtained from spectrum of mixture of isomers), δ, ppm: 35.5 (CH_2_), 41.8 (CH), 47.1 (C(CN)_2_), 82.7 (C≡), 88.1 (C≡), 113.1 (CN), 121.4, 132.13, 133.9, 136.1, 194.3 (CO). ^13^C NMR, **4b** (selected signals, obtained from spectrum of mixture of isomers), δ, ppm: 34.6 (CH_2_), 40.6 (CH), 47.3 (C(CN)_2_), 83.0 (C≡), 88.0 (C≡), 113.2 (CN), 113.3 (CN), 121.3, 132.08, 134.0, 136.0, 194.4 (CO). ^13^C NMR, (signals of mixture of isomers) δ, ppm: 126.7, 128.29, 128.33, 128.4, 128.89, 128.91, 129.06, 129.11, 142.7, 144.3. IR (KBr): 1691 cm^−1^ (CO), 2238 cm^−1^ (CN, C≡C). HRMS (ESI-TOF) *m/z*: [M + Na]^+^ Calcd for C_37_H_26_N_2_O_2_Na 553.1892; Found 553.1886.

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
