# Peer review of "Stereoselective Synthesis of Multisubstituted Cyclohexanes by Reaction of Conjugated Enynones with Malononitrile in the Presence of LDA"

_molecules, 2020, doi:10.3390/molecules25245920_

Round 1
Reviewer 1 Report
This article describes the synthesis of 8 polysubstituted cyclohexanes by a reaction between enynones and malononitrile using LDA as a base.
The originality of this study is very low (see ref 18-22) and its practical interest is limited by the poor yields.
Remarks:
- The text contains typos and some grammatical errors.
- In many reported NMR spectra, signals not corresponding to the described molecule are observed. The authors should improve the purity of their compounds.
- page 2, line 68-69. “only one isomer” should be “only one pair of enantiomers” and “stereoselectivity” should be “diastereoselectivity”.
- Table 1: The number of equivalent of LDA should be reported.
- Scheme 2: Where the proton comes from to go from A to 3? The authors should write “work up” above the arrow (which should be non-reversible) instead and clearly show that B does not come from 3 but directly from A (equilibrium arrows). “H+” should be changed by “work up” above the arrows going to 4 and 2 as well.
- Page 7, paragraph below scheme 2: it would be interesting to report relative free energy values of the different diastereomers.
- In section 4.2, the nature of the solvent modelled should be provided.
- The authors have optimized the structure of the 16 isomers of cyclohexane 2a. These 16 isomers are in fact 8 pairs of enantiomers. Indeed, 2a1 is the enantiomer of 2a12, 2a2 is the enantiomer of 2a10 etc. Since enantiomers have identical physical properties (including free energy) there is no point of optimizing two enantiomers. By the way, how the authors can find differences in relative free energy between enantiomers, see for instance between 2a1 and 2a12 → 86 kJ/mol, between 2a2 and 2a10 → 80 kJ/mol etc etc
Author Response
Reviewer 1
This article describes the synthesis of 8 polysubstituted cyclohexanes by a reaction between enynones and malononitrile using LDA as a base.
The originality of this study is very low (see ref 18-22) and its practical interest is limited by the poor yields.
Remarks:
- The text contains typos and some grammatical errors.
Answer:
English language was polished.
- In many reported NMR spectra, signals not corresponding to the described molecule are observed. The authors should improve the purity of their compounds.
Answer:
Indeed, some NMR spectra contain signal of impurities, such as, petroleum ether, water, etc. In some cases, traces of other diastereomers may be observed, but amount of these minor products were undetectable.
- page 2, line 68-69. “only one isomer” should be “only one pair of enantiomers” and “stereoselectivity” should be “diastereoselectivity”.
Answer:
These changes were done.
- Table 1: The number of equivalent of LDA should be reported.
Answer:
The number of equivalents of LDA was added to the Scheme in Table 1: LDA (1 equiv.).
- Scheme 2: Where the proton comes from to go from A to 3? The authors should write “work up” above the arrow (which should be non-reversible) instead and clearly show that B does not come from 3 but directly from A (equilibrium arrows). “H+” should be changed by “work up” above the arrows going to 4 and 2 as well.
Answer:
Scheme 2 and the corresponding text was changed according to reviewer’s comments.
- Page 7, paragraph below scheme 2: it would be interesting to report relative free energy values of the different diastereomers.
Answer:
The following phrase was added^
“Difference in relative energies of diastereomers reaches up to 49 kJ/mol (see Supporting Information).
- In section 4.2, the nature of the solvent modelled should be provided.
Answer:
Solvent was water. It has been added to section 4.2.
- The authors have optimized the structure of the 16 isomers of cyclohexane 2a. These 16 isomers are in fact 8 pairs of enantiomers. Indeed, 2a1 is the enantiomer of 2a12, 2a2 is the enantiomer of 2a10 etc. Since enantiomers have identical physical properties (including free energy) there is no point of optimizing two enantiomers. By the way, how the authors can find differences in relative free energy between enantiomers, see for instance between 2a1 and 2a12 → 86 kJ/mol, between 2a2 and 2a10 → 80 kJ/mol etc etc
Answer:
We did calculation again. Correct data for the most stable conformers are given in SI.
Reviewer 2 Report
The paper of Iguskina et al. Is an interesting one and worth publishing in Molecules. However, there are several thing which should be corrected and clarified. Moreover English also should be improved.
My remarks:
Row 39: Catch/isolate, understand/elucidate
Row 65: We note, that in
Row 73:ethynyl-substituted cyclohexanes is presented very poorly??? Limitedly occuring topic, limitedly discussed
Row: 73-75: However these compounds may have remarkable practical importance as photoluminescent complex ligands, and as useful building bloks for densely substituted cyclohexanes.
Row 161: Conclusions: The conclusion is short and meaningless, please improve it.
Row 162: in the presence of strong base as LDA...
Row 203 : washed/quenched
Row 348: the temperature of liquid nitrogen bath is not -70C, it is -195 C (Do you mix it EtOAc?)
Row 350: 1a bold
Row 355: 3a, 4ab bold
References: As far as I know the policy of this journal requires to include the full title of papers, starting and ending pages also.
Substantial remarks: I am not convinced that only one enantiomer pair is formed (by the way, it is not clear me, whether the substance authors received is one enentiomer or enantiomer pair (racemate) and it is should be indicated in the nomenclature of compounds), yields are sometimes low 17-18%. May be mixture of diastereomers are formed, but only one crystallizes out. Please indicate these in text and/or give experimental proof.
Author Response
Reviewer 2
The paper of Iguskina et al. Is an interesting one and worth publishing in Molecules. However, there are several thing which should be corrected and clarified. Moreover English also should be improved.
My remarks:
Row 39: Catch/isolate, understand/elucidate
Answer:
These changes were done.
Row 65: We note, that in
Answer:
These changes were done.
Row 73:ethynyl-substituted cyclohexanes is presented very poorly??? Limitedly occuring topic, limitedly discussed
Answer:
These changes were done.
Row: 73-75: However these compounds may have remarkable practical importance as photoluminescent complex ligands, and as useful building bloks for densely substituted cyclohexanes.
Answer:
These changes were done.
Row 161: Conclusions: The conclusion is short and meaningless, please improve it.
Answer:
Conclusion was changed.
Row 162: in the presence of strong base as LDA...
Answer:
These changes were done.
Row 203 : washed/quenched
Answer:
These changes were done.
Row 348: the temperature of liquid nitrogen bath is not -70C, it is -195 C (Do you mix it EtOAc?)
Answer:
It was a liquid nitrogen-acetone bath. Now, it is corrected in this procedure.
Row 350: 1a bold
Answer:
These changes were done.
Row 355: 3a, 4ab bold
Answer:
These changes were done.
References: As far as I know the policy of this journal requires to include the full title of papers, starting and ending pages also.
Substantial remarks: I am not convinced that only one enantiomer pair is formed (by the way, it is not clear me, whether the substance authors received is one enentiomer or enantiomer pair (racemate) and it is should be indicated in the nomenclature of compounds), yields are sometimes low 17-18%. May be mixture of diastereomers are formed, but only one crystallizes out. Please indicate these in text and/or give experimental proof.
Answer:
We obtained racemates. Other diastereomers were not formed, since we collected all compounds after chromatographic separation.
This statement was added in the 1-st paragraph of Result and Discussion Section.
Stereochemical descriptors R, S were added to names of compounds 2.
Reviewer 3 Report
In this manuscript, authors describe a practical method for the preparation of highly substituted cyclohexanol derivatives based on Michael addition of malononitrile to enynones followed by annulation stereo-selectively. It is an interesting work to prepare such compounds by using enynone as the substrates with the yne functionality intact during the reaction. Despite of a somewhat limited scope on functional groups in the molecules, this study provides the knowledge about the stereochemistry outcome of the final product. Authors have determined the stereochemistry of compound by single crystal structural analysis, but not much words on its pathway leading to this outcome. Considering the readily availability of the substrates and operational simplicity, this method should be of interest to the synthetic chemist.
After authors address the following points, I believe it is appropriate for a publication in this journal.
- In Figure 2, please re-draw the suitable chair conformation of the compound, for example, Ha and Hb should be in parallel arrangement, so are tow CCAr1.
- Please check spectra of compounds 2 carefully again. As indicated in Figure 2, it says that proton Hd appears to singlet. I do not think it is correct.
- Table 1 entry 4 shows that 4a/4b (diastereoisomers) in 1.5:1. Did authors check this ratio by warming this mixture? or even follow the stereo-chemical outcome of the final cyclohexanols from this batch?
- Although the stereochemistry of 2 is thermodynamic stable, in scheme 2, elaboration of stereochemistry of the intermediates and products is suitable.
- Since it is a stereoselective reaction, authors should include this point in the conclusion.
Author Response
Reviewer 3
In this manuscript, authors describe a practical method for the preparation of highly substituted cyclohexanol derivatives based on Michael addition of malononitrile to enynones followed by annulation stereo-selectively. It is an interesting work to prepare such compounds by using enynone as the substrates with the yne functionality intact during the reaction. Despite of a somewhat limited scope on functional groups in the molecules, this study provides the knowledge about the stereochemistry outcome of the final product. Authors have determined the stereochemistry of compound by single crystal structural analysis, but not much words on its pathway leading to this outcome. Considering the readily availability of the substrates and operational simplicity, this method should be of interest to the synthetic chemist.
After authors address the following points, I believe it is appropriate for a publication in this journal.
- In Figure 2, please re-draw the suitable chair conformation of the compound, for example, Ha and Hb should be in parallel arrangement, so are tow CCAr1.
Answer:
Chair conformation was re-drawn.
- Please check spectra of compounds 2 carefully again. As indicated in Figure 2, it says that proton Hd appears to singlet. I do not think it is correct.
Answer:
Signal Hd is a widened singlet due to low spin-spin interaction constant of this proton with its neighbour protons.
- Table 1 entry 4 shows that 4a/4b (diastereoisomers) in 1.5:1. Did authors check this ratio by warming this mixture? or even follow the stereo-chemical outcome of the final cyclohexanols from this batch?
Answer:
At the warming of reaction mixture, reaction goes dipper to compound 2.
- Although the stereochemistry of 2 is thermodynamic stable, in scheme 2, elaboration of stereochemistry of the intermediates and products is suitable.
Answer:
Structures in Scheme 2 were changed to show elaboration of stereochemistry.
- Since it is a stereoselective reaction, authors should include this point in the conclusion.
Answer:
Conclusion was changed.
Round 2
Reviewer 1 Report
The authors have taken most of the remarks into account. In my opinion, the purity of several compounds is still to be improved in order to be able to report isolated yields.
Reviewer 3 Report
In this manuscript, authors describe a practical method for the preparation of highly substituted cyclohexanol derivatives stereo-selectively. It is an interesting work. As mentioned earlier, authors should address more detail about the stereochemistry. Now, they have revised this issue accordingly. I think the present work is suitable for a publication in this journal.